# NMR-Based Metabolomics: A New Paradigm to Unravel Defense-Related Metabolites in Insect-Resistant Cotton Variety through Different Multivariate Data Analysis Approaches

**DOI:** 10.3390/molecules28041763

**Published:** 2023-02-13

**Authors:** Anam Amin Shami, Muhammad Tayyab Akhtar, Muhammad Waseem Mumtaz, Hamid Mukhtar, Amna Tahir, Syed Shahzad-ul-Hussan, Safee Ullah Chaudhary, Bushra Muneer, Hafsa Iftikhar, Marios Neophytou

**Affiliations:** 1Department of Biotechnology, University of Central Punjab, Lahore 54000, Pakistan; 2Institute of Industrial Biotechnology, GC University Lahore, Lahore 54000, Pakistan; 3Department of Chemistry, University of Gujrat, Gujrat 50700, Pakistan; 4Biomedical Informatics and Engineering Research Laboratory, Department of Life Sciences, Syed Babar Ali School of Science and Engineering, Lahore University of Management Sciences, Lahore 54792, Pakistan; 5Department of Biology, Syed Babar Ali School of Science and Engineering, Lahore University of Management Sciences, Lahore 54792, Pakistan; 6KAUST Solar Center (KSC), King Abdullah University of Science and Technology (KAUST), Thuwal 23955-6900, Saudi Arabia

**Keywords:** metabolomics, NMR, cotton, defense related metabolites, multivariate data analysis

## Abstract

Cotton (*Gossypium hirsutum*) is an economically important crop and is widely cultivated around the globe. However, the major problem of cotton is its high vulnerability to biotic and abiotic stresses. It has been around three decades since the cotton plant was genetically engineered with genes encoding insecticidal proteins (mainly Cry proteins) with an aim to protect it against insect attack. Several studies have been reported on the impact of these genes on cotton production and fiber quality. However, the metabolites responsible for conferring resistance in genetically modified cotton need to be explored. The current work aims to unveil the key metabolites responsible for insect resistance in Bt cotton and also compare the conventional multivariate analysis methods with deep learning approaches to perform clustering analysis. We aim to unveil the marker compounds which are responsible for inducing insect resistance in cotton plants. For this purpose, we employed 1H-NMR spectroscopy to perform metabolite profiling of Bt and non-Bt cotton varieties, and a total of 42 different metabolites were identified in cotton plants. In cluster analysis, deep learning approaches (linear discriminant analysis (LDA) and neural networks) showed better separation among cotton varieties compared to conventional methods (principal component analysis (PCA) and orthogonal partial least square discriminant analysis (OPLSDA)). The key metabolites responsible for inter-class separation were terpinolene, α-ketoglutaric acid, aspartic acid, stigmasterol, fructose, maltose, arabinose, xylulose, cinnamic acid, malic acid, valine, nonanoic acid, citrulline, and shikimic acid. The metabolites which regulated differently with the level of significance *p* < 0.001 amongst different cotton varieties belonged to the tricarboxylic acid cycle (TCA), Shikimic acid, and phenylpropanoid pathways. Our analyses underscore a biosignature of metabolites that might involve in inducing insect resistance in Bt cotton. Moreover, novel evidence from our study could be used in the metabolic engineering of these biological pathways to improve the resilience of Bt cotton against insect/pest attacks. Lastly, our findings are also in complete support of employing deep machine learning algorithms as a useful tool in metabolomics studies.

## 1. Introduction

The development of plants with improved agronomic traits has revolutionized the cash crop (e.g., wheat, cotton, and rice) production through genetic engineering. Moreover, over the years genetic engineering has proven to play a substantial role in the development of crops with enhanced traits yielding better productivity and resistant characteristics against insects and herbicides. Resultantly, these genetically engineered crops generate significant revenues and yields [1,2].

Cotton (*Gossypium hirsutum*) is one of the major cash crops in the world. It is included among the important agricultural contributors towards economic growth for both developed and underdeveloped countries, and is found to be ranked amongst the major agricultural contributors which produce a significant impact on the economy of both developed and middle-income countries. However, according to an evaluation. About 87% of cotton is provided by developing countries [3]. Further, a retrospective study showed that cotton being a major stakeholder in the world’s economy brought about an economic impact of $600 billion in the world’s leading textile industries [4]. In addition to the economic impact, cotton shows a high vulnerability to both biotic and abiotic stresses, as reported by numerous studies about the key role of these stress factors as key players in the deterioration of cotton production [5,6]. For this very reason, cotton was genetically engineered with genes encoding insecticidal proteins. The commercialization of Bt cotton has revolutionized the cotton industry, it has raised the farmer’s benefit by 261–438 US$ per hectare [7]. Therefore, Bt-engineered cotton is safeguarding crops against major devastators like *Helicoverpa armigera*, pink bollworm, and diamond black moth [8]. Studies have been reported on the impact of these genes on cotton production and fiber quality [9]. However, the metabolites responsible for conferring resistance in genetically modified cotton need to be explored.

Plants produce a large number of metabolites of heterogeneous nature, which play important roles in plant growth, development, and their response to the external environment [10]. These small molecules (metabolites) are valuable nutritional sources not only for plants but also for humans and livestock. According to an estimate, nearly 200,000 compounds have been found in the plant kingdom and out of 200,000, only 50,000 metabolites are known so far [11]. Plant metabolites, particularly secondary metabolites, play a prominent role in plants’ defense mechanisms. Secondary metabolites include alkaloids, terpenoids, flavonoids, isoprenoids, phenylpropanoids, and nitrogen- and sulfur-based phenolic compounds [12]. Summarily, these phytochemicals serve as a preliminary line of defense against biotic stresses [13]. Terpenoids such as iridoids and benzoxazinoids play an active role in plant defense [14]. Gossypol, a sesquiterpenoid aldehyde was found to be phytoalexin in nature. Similarly, hemigossypolene showed anti-fungal activities against *A. flavus* [15]. Thus, the diversity of plant metabolites and their complicated regulatory mechanisms have stressed upon the need to explore them in depth. Consequently, making plant metabolomics an emerging field of science for researchers to explore different tangents in order to get a dynamic snapshot of their underlying biochemical pathways, identification of plant disease biomarkers, and studying plants’ response to biotic and abiotic stresses [16].

Plant metabolomics refers to the qualitative and quantitative study of small molecules (<1500 Da). Therefore, this field of science is equipped with the provision of highly accurate information related to the physiological state of an organism [17]. Different analytical techniques like gas chromatography–mass spectrometry (GC-MS), liquid chromatography–mass spectrometry (LC-MS), nuclear magnetic resonance spectroscopy (NMR), liquid chromatography nuclear magnetic resonance (LC-NMR), and liquid chromatography solid phase extraction nuclear magnetic resonance (LC-SPE-NMR) have been employed in plant metabolomics [18,19]. LCMS is a more sensitive approach than GCMS and NMR though the major disadvantage of this platform is ion suppression. Therefore, NMR is used as a complementary technique to GCMS and LCMS [20]. NMR spectroscopy is robust, non-invasive, and highly reproducible including simple sample preparation [21]. It is an ideal technique to study real-time metabolite profiling in living systems [22].

Previously, Ren et al. [23] examined the metabolic profile of wild-type and transgenic Arabidopsis plants through NMR spectroscopy. Similar work was done to study host plant resistance against thrips by using ^1^H-NMR spectroscopy [24]. Likewise, Becerra–Martinez et al. [25] deployed NMR-based metabolomics to investigate differences in the metabolic composition of transgenic tobacco plants expressing sequisterpene cyclase. Herein, the present study we have used NMR spectroscopy for the metabolomic analysis of Bt and non-Bt cotton varieties. The present work aimed to unveil marker compounds responsible for insect resistance in Bt cotton plants. Furthermore, it explores the impact of Bt genes on the expression level of the key metabolites.

## 2. Results

### 2.1. ^1^H-NMR Identification of Metabolites in Cotton Varieties

^1^H-NMR was used to unravel the metabolic dynamics between FH-142 (Bt cotton), Mac-07 (pest resistant), and 496 (wild) cotton varieties. The representative ^1^H-NMR spectra are shown in Figure 1. The identification of metabolites was done by consulting previously published literature [26,27,28] and metabolite databases [29,30]. The detailed metabolites along with their multiplicity pattern are tabulated in Table 1. The assignment of metabolites was validated through 2D J-resolved experiment (Appendix A). The ^1^H-NMR spectra of different *G. hirsutum* varieties can be divided into three spectral regions i.e., a high field or aliphatic (δ 0.5–3.0 ppm), saccharide (δ 3.0–5.5 ppm), and aromatic region (δ 5.5–10.00 ppm). The major constituents identified were amino acids, organic acids, lipids, terpenoids, and flavonoids. In the aliphatic region, characteristic signals of four non-essential amino acids. Namely, alanine, γ-amino butyric acid (GABA), asparagine, citrulline, and one essential-amino acid (valine) were identified. Additionally, successful detection of succinic acid at δ 2.45 (s), aspartic acid (δ 2.68, dd), α-Ketoglutaric acid (2.43, t), and terpenoids (limonene and terpinolene) were also noted. Regardless of overlapping in the carbohydrate region, anomeric proton signals of arabinose, fructose, xylulose, maltose, and sucrose were observed. Likewise, the anomeric proton signal of sugar moieties of choline and sycloinositol was also pointed at δ 3.20 (s) and δ 3.21 (s), respectively. The diagnostic signals of organic acids namely aconitic acid (δ 3.43, d) shikimic acid (δ 4.01), and tartaric acid (δ 4.35) were also observed. In the low field of ^1^H-NMR spectra (Figure 1), characteristic signals of uridine (δ 5.95, d) were detected. The signals for phenylpropanoid derivative (cinnamic acid) were found at δ 6.52 (d) and δ 7.60 (m). The 2H signal of fumarate was identified at δ 6.5 (s). However, the characteristic signals of dibutyl phthalate (δ 7.7, s), formate (δ 8.47, s), and trigonelline (δ 9.14, s) were also annotated. The visual inspection revealed marginal metabolic differences between the wild (496) and pest-resistant (FH-142 and MAC-07) varieties of cotton. The most prominent difference observed among cotton varieties is the presence of stigmasterol in non-transgenic pest-resistant varieties (MAC-07). Moreover, the signal intensity of valine, limonene, succinic acid, terpinolene, γ-aminobutyric acid, sycloinositol, proline, shikimic acid, chlorogenic acid, ferulic acid, arabinose, xylulose, trigonelline, and formate was found higher in FH-142. In addition to visual inspection, we performed a visual representative analysis (heat map) of the NMR data to find out metabolic differences among cotton varieties.

### 2.2. Heat map Analysis

Heat maps assisted in visualizing the variations in the metabolic contents with respect to three different cotton varieties. Further, in order to attain a clear picture of clustering between the metabolites as well as different types of cotton, a data refinement step was performed, where a single data encounter represented the geometric mean of the 6 replicates of each cotton variety calculated for each expression of metabolite. Figure 2 illustrates a heat map along with dendrograms showing clustering along both axes, i.e., between cotton varieties and the level of expression of various metabolites. To understand the expression level, the red and blue color code was used where a red color represents a higher expression level, while blue color depicts a lower expression and white signifies no change. As a result, Figure 2 evidently distinguishes the FH-142 variety to be different from MAC-07 and 496. This could be validated by the outcome in form of a heat map and dendrogram which is illustrating an entirely different pattern of expression for FH-142. Furthermore, a separate branching from the cluster of MAC-07 and 496 varieties reveals that these two varieties share a somewhat similar metabolite profile in comparison to FH-142. Summarily, primary metabolites such as tryptophan, tyrosine, arabinose, citrulline, maltose, fumarate, and GABA were highly expressed in the Bt cotton variety (FH-142). Likewise, the secondary metabolites cinnamic acid, succinic acid, stigmasterol, limonene, linoleic acid, and E-β-ocimene were also found to have increased expression in Bt cotton. This inferred that induction of Cry genes in cotton variety resulted in increased production of primary and secondary metabolites in cotton plants.

### 2.3. Multivariate Data Analysis

The variation in the metabolite content of cotton varieties was also evaluated by using multivariate data analysis (MvDA). The principal component analysis (PCA) was applied to apprehend the clustering features of three different cotton varieties. In PCA, five components contributed 84% total variance. PC1 explains 53% variation while PC2 accounts for 13% of the variation in data. It can be seen in Figure 3 that PCA was unable to show any meaningful separation among the data. The PCA results did not exhibit notable differences among samples so we applied Orthogonal partial latent square discriminant analysis (OPLSDA) to the data.

The OPLSDA model shows a complete discrimination among the cotton varieties. The model depicts high goodness of fit and low predictability with an R2X value of 0.888 and a Q2 value of 0.362. The model was further validated through a 100 permutations test and a Q2 value of −0.361 was found as shown in Figure 4B. The OPLSDA score plot shown in Figure 4A clearly separates FH-142 (Bt cotton) from MAC-07 and 496. FH-142 was separated by OPLS1 and OPLS2 from the other two varieties, whereas Mac-07 and 496 were projected together on the positive side of OPLS1. A detailed examination of the loading plot (Figure 4C) shows that terpinolene, α-ketoglutaric acid, aspartic acid, cinnamic acid, stigmasterol, fructose, maltose, arabinose, xylulose, cinnamic acid, and Shikimic acid were responsible for separating FH-142 from MAC-07 and 496. Consequently, malic acid and valine correspond more to 496 and MAC-07.

Although OPLSDA clearly separated FH-142 from MAC-07 and 496, MAC-07 and 496 were unable to be discriminated against. Hence, we performed the latest deep learning approaches such as LDA and neural networks on NMR data as a feature reduction approach. Initially, LDA failed to discriminate against the cotton varieties and resultantly most of the predictors given were in the unidentified region as a solution. We further filtered the data by excluding the unidentified region and performed LDA and neural network analyses. Consequently, there was a notable improvement in the results, as shown in Figure 5A. The score plot of LDA accounted for a 100% group classification among the cotton varieties. The discriminant analysis produced fourteen potential predictors classified the types of cotton at a significance level of 0.001, as elucidated in Figure 5B. The Fisher’s discriminant functions revealed that nonanoic acid, valine, alanine, citrulline, and arginine were major contributors in distinguishing between cotton varieties. Coefficients with the highest scores were attributed more toward the classification of a particular group. The classification results as presented in Figure 5C showed that 100% of the originally grouped data was correctly classified and showed a 100% predicted group membership. The fourteen significantly shortlisted predictors from LDA were further subjected to multilayer perceptron neural network analysis, which is a supervised deep learning approach deployed to test the efficiency of the data. The data consisting of independent variables were randomly divided into training and testing sets. The classification results (Figure 6A) from the multilayer perceptron model provided 100% correct prediction calculations for the training and testing data sets. The individual importance of variables in the prediction of a particular variety is elucidated in Figure 6B,C, which showed that malic acid, arginine, citrulline, and valine were the most important metabolites in distinguishing each cotton variety in the present model. Our results showed that there is concordance in the metabolites identified by conventional methods (PCA and OPLSDA) and deep learning methods (LDA and neural networks). However, OPLSDA has given more discriminating metabolites than LDA but LDA showed a clear separation between cotton varieties *on the basis of 14 predictor metabolites, depicting clear discrimination of cotton variety to a* specific *class.* Moreover, relative quantification results (Table and graphs (Appendix A)) also displayed a comparative level of differentiating metabolites among the studied cotton groups (Appendix A).

### 2.4. Metabolic Pathway Analysis

The annotated metabolites were characterized under general biochemical pathways such as tricarboxylic acid cycle (TCA), Shikimic acid pathway, glutamine synthetase/synthase (GS/GOGAT) cycle, and amino acid biosynthetic metabolism based on the search results in KEGG. The induction of the Cry gene regulated the concentration of organic acids, terpenoids, amino acids, and flavonoids. In the TCA cycle, the concentration of aconitic acid, α-ketoglutaric acid, succinic acid, and fumarate has increased in FH-142 (Bt cotton) as highlighted in green color in Figure 7A. The expression of Cry genes was shown to cause an incremental effect in their expression on a number of major intermediates of Shikimic acid pathways. The levels of Shikimic acid and quinic acid were significantly enhanced in FH-142 compared to other varieties. Furthermore, the upregulation of tryptophan (0.103) and tyrosine (0.0085) was observed in the FH cotton variety (Appendix A). Similarly, the content of cinnamic acid and ferulic acid was also upregulated in FH-142. As these metabolites, along with shikimic acid, are important intermediates of plants’ secondary metabolism, the enhanced production of these compounds would result in a better defense mechanism in Bt cotton. Figure 7B illustrates the biological schema of the secondary metabolites and their flow of interrelationships. In GS/GOGAT biosynthetic pathway, transamination-related metabolites were also upregulated in FH-142 (Bt cotton).

## 3. Discussion

Recently, plant metabolomics has been extensively used to assess metabolic perturbations in plants caused by biotic or abiotic stresses [31] as well as for risk assessment of genetically modified crops [32]. **Thereby**, the present work was majorly focused on the identification of biomarkers responsible for insect resistance in pest-resistant cotton varieties. The metabolite characterization results depicted that cotton varieties are enriched in bioactive compounds. Among the identified compounds, the amount of cinnamic acid, E-β-ocimene, tryptophan, valine, uridine, dibutyl phthalate, GABA, succinic acid, fumarate, shikimic acid, arabinose, xylulose, trigonelline, glycine, di-allylic methylene, tyrosine, and malic acid was found to be high in FH-142 (Bt cotton) as compared to other varieties. Levande et al. [33] employed capillary electrophoresis to compare metabolic profiles of three Bt transgenic maize varieties with wild maize plants, the results revealed a significant difference in overall metabolite content. Likewise, Ning et al. [34] used GC–MS and UPLC–MS/MS to examine the cambium metabolomes of multi-gene stress resistant transgenic lines. The study concluded that a notable difference in the relative abundances of sucrose, arginine, uridine diphosphate glucose, glutamate, and catechol was observed in multi-gene stress-resistant transgenic lines of popular plants in comparison to non-transgenic ones. Further, we can state that the findings of our metabolomic analyses are in agreement with the past studies.

Amino acids are key products of primary metabolism, they are involved in multiple functions in plants such as growth promoters, cell wall biosynthesis, osmoregulator, and intermediates of secondary metabolites [27]. Previous data have shown variation in amino acid content among transgenic lines as compared to wild type [34]. In the present study, a significant increase in the amount of valine, GABA, tryptophan, tyrosine, glycine, and arabinose was observed in FH-142. Additionally, aromatic amino acids such as tyrosine and tryptophan are not only the major components of plant proteins, but are also involved in the upregulation of growth hormones and secondary metabolites [27]. Tyrosine is a precursor of various secondary metabolites that serve as attractants and defensive compounds [35]. Tryptophan, an essential amino acid, plays an important role in the biosynthesis of precursors involved in plant growth, defense against biotic and abiotic stresses, and plant-insect interaction [36]. Valine is involved in the biosynthesis of glucosinolates which has a deterring effect on micro-organisms and herbivory [37]. A previous study showed that by exogenous application of amino acids on flowers of Bt cotton the concentration of Bt proteins could be increased [38], therefore, proposing that the concentration of Bt proteins could be increased with the upregulation of amino acids. GABA, a non-protein amino acid act as a signaling molecule in plants [39]. It works either by having a direct inhibitory effect by regulating a defense mechanism or a combination of both [39]. Shelp et al. [40] reported the indirect role of GABA in plant growth as well as the regulation of defense mechanisms against biotic and abiotic stresses. Scholz et al. [41] investigated the impact of insect feeding on the mutants of Arabidopsis thaliana. The results demonstrated that accumulation of GABA occurred upon insect infestation. Furthermore, a decline in the development of *Spodoptera littoralis* larvae was observed when fed on GABA enriched artificial diet. Bown et al. [42] observed that non-wounding insects crawling on leaves induced the production of GABA, thus inferring the role of GABA in the defense mechanism of plants. The enhanced production of GABA in Bt cotton could possibly be responsible for conferring resistance against *Helicoverpa armigera*, pink bollworm, and diamond black moth.

Carbohydrates are major energy bearers in plants [43]. Carbohydrates such as sucrose, glucose, fructose, and raffinose are notable compatible solutes in plants [44]. In our work, elevation in the concentration of arabinose and xylulose was observed in Bt cotton. Arabinose, which is a constituent of plant cell walls, plays a pivotal role in the synthesis of cell wall intermediates, flavonoids, and signaling peptides [45]. Zayed [46] studied the role of arabinose in plants under salt stress. He found that arabinose-based glycoproteins help plants in tolerating salt stress. Therefore, a high concentration of arabinose in leaves of FH-142 might play an important role in tolerating abiotic stress.

Among organic acids, fumarate, malic acid, succinic acid, shikimic acid, and malic acid were found in high concentrations in FH-142. These metabolites are intermediates of primary and secondary metabolism. Fumarate, malic acid, and succinate are intermediates of the tricarboxylic acid cycle (TCA), and are also involved in regulating pH in plants [47]. A study conducted by Zhou et al. [48] demonstrated irregularity in the levels of TCA intermediates in Bt-transgenic rice grown under insecticidal stress. After an early decrease, a significant increase in the concentration of TCA intermediates was reported. These findings are in good agreement with our results. Further, Fahnenstich et al. [49] proposed that malate and fumarate play key roles in the primary metabolism of *A. thaliana*. Reactive oxygen species (ROS) are normally produced in plants. An increase in the concentration of ROS species occurring during the stress conditions leads to oxidative stress [50], which resultantly assists in the release of hydroxyl radicals which damages plant macromolecules especially proteins [51]. However, Succinic acid and malic acid have scavenging effects on ROS species [52]. The upregulation of these metabolites possibly protects FH-142 against abiotic stress. Shikimic acid is an indispensable component of the Shikimate and phenylpropanoid pathways [53]. The elevated level of shikimic acid is the result of the biosynthesis of secondary metabolites which are involved in the plant’s defense mechanisms such as flavonoids and lignins [48]. Thus, a high level of shikimate might lead to the upregulation of secondary metabolites in FH-142 which can lead to better defense against pest attacks.

In the current study, a few terpenoids including limonene, terpinolene, stigmasterol, and E-β-ocimene were also identified. E-β-ocimene, a monoterpene, is known to help plants in pollination. β-ocimene is a major component of floral scents [54]. In addition to playing the role of pollinator attractor, E-β-ocimene was found to be involved in direct insect defense in *M. truncatula* [55]. Kang et al. [56] proposed that β-ocimene was responsible for activating the defense mechanism of Chinese cabbage against *M. persicae*. In cotton plants, β-ocimene was found to be synthesized in response to insect attacks [6]. We have also found a high level of E-β-ocimene in FH-142, which may also be responsible for inducing insect resistance in Bt cotton.

Plants use nucleotides to induce defense mechanisms against predators and pathogens [57]. We observed an elevated level of uridine in FH-142, which can also be an important factor in inducing the cotton plant’s defense mechanism. Furthermore, we also observed different levels of cinnamic acid in pest-resistant varieties as compared to non-pest-resistant varieties.

Multivariate data analysis (MvDA) has extensively been used in metabolomic studies [58]. It is primarily used to process massive data obtained through analytical approaches, albeit yielding useful information [59]. The two most commonly used MvDA platforms are the principal component analysis (PCA) and the partial least square regression (PLS) method [60]. PCA is an unsupervised approach, often used as a conventional method in plant metabolomics [61]. Although, PCA is a beneficial platform in MvD; however, occasionally it fails to analyze data obtained from multifactorial experiments [62]. In the current work, PCA was found unable to differentiate the cotton varieties. Therefore, we applied different algorithms like OPLSDA, LDA, and neural networks to differentiate cotton varieties. OPLSDA has given more discriminating metabolites, in comparison to LDA which presented a model with lesser predictors. However, LDA showed a clear segregation between the cotton varieties compared to OPLSDA. The difference in the discriminating metabolites given by OPLSDA and LDA is mainly due to the dimensions in which these tools project the data for feature reduction. LDA projects data in a low dimensional space i.e., K-1 [63]. While PCA projects data in K- dimensions [64]. Balakrishnama and Ganapathiraju, Ref. [65] reported that PCA does more feature classification, while LDA does more data separation which is in accordance with our results. Likewise, previously reported data by Alves et al. [66] applied nine different algorithms to find the best identification tool for fibromyalgia. They inferred that SPA-LDA is a reliable tool in the clinical diagnosis of fibromyalgia. It concluded that NMR-based metabolomics conjoined with multivariate data analysis. In particular, LDA is a powerful platform in differentiating plant samples. Moreover, the elevated levels of defensive metabolites in FH-142 might be responsible for conferring resistance against insect herbivory.

## 4. Materials and Methods

### 4.1. Chemicals

All the chemicals and solvents utilized in the present work were of analytical grade and purchased from Cambridge Isotope Laboratories and Acros. The chemicals used were included deutrium oxide (D2O) (Cat-No: 14D-099), sodium deuterium oxide (NaOD) (Cat-No.: DLM-45-PK), methanol-d4 (Cat No. DLM-24-PK), 2,2,3,3-D4 (D,98%) sodium-3-trimethylsilylpropionate (TMSP) (Cat-No: I-18625), potassium dihydrogen phosphate (KH2PO4) (Cat No. AC424200250), and liquid nitrogen.

### 4.2. Sample Collection

Three different varieties of cotton plant namely MAC-07 (non-transgenic insect = resistant American variety), FH-142 (transgenic insect-resistant variety having Cry proteins, Bt cotton), and 496 (non-insect resistant variety) were procured from the Institute of Agriculture sciences, Punjab University Lahore. The leaf samples from six replicates of each variety were harvested 6 months after the sowing of cotton seeds, labeled, and plunged directly into liquid nitrogen. The samples were quenched to stop enzymatic degradation. Samples were then brought to the research lab of IIB, GCU Lahore. Afterward, the leaves were cryogenically dried using a mortar and pestle, and lyophilized for 48–72 h. The lyophilized plant samples were stored in a −80 °C freezer (Model: MDF-594, Ultra-low, SANYO, Osaka, Japan) till further use.

### 4.3. NMR Sample Preparation

The plant extract was prepared by following the previously reported protocol (Hussain et al., 2018 [17]) with minor modifications. A sample of 50 mg lyophilized plant material was transferred to an Eppendorf tube, to which methanol d4 (500 µL) and 500 µL of KH_2_PO_4_ buffer (in D2O containing 0.01% TMSP, pH 6.0) were added. To ensure proper mixing of the constituents, the mixture was vortexed for 1 min followed by ultrasonication for 20 min. Later, the extract was centrifuged (SIGMA laboratory centrifuges 3K30) for 5 min at 13,000 rpm at 4 °C. After centrifugation, the supernatant of each sample (600 µL) was transferred to a 5 mm NMR tube for NMR analysis.

### 4.4. NMR Acquisition

NMR acquisition was performed by following the parameters as mentioned in (Hussain et al., 2018 [17]) with few modifications. ^1^H-NMR spectra were obtained by using Avance neo 600 MHz spectrophotometer. Methanol-d4 was used as an internal lock. The spectra were recorded with 128 scans and 64K data points using Bruker’s pulse program “ZG” without solvent suppression. The spectra were Fourier transformed with line broadening (Lb = 0.3). However, no zero filling was added.

### 4.5. Data Processing

The ^1^H-NMR spectra were referenced to TMSP (at δ 0.0 ppm), manually phase and baseline corrected by using MestreNova (14.0 version software). Later, all the spectra were reduced to the ASCII file prior to MvDA. The spectral intensities were normalized to the total area. The spectral region from δ 0.5–9.0 ppm was bucketed into bins of equal width i.e., 0.01 thus yielding 995 variables for each spectrum. The residual water (δ 4.7–4.8) and solvent peak (δ 3.30) was excluded from the data. The resultant file was imported to Simca P (14.0 version) for multivariate data analysis.

### 4.6. Multivariate Data Analysis (MvDA)

Heat map analysis was performed in order to have a systemic overview and clustering of metabolic distribution, by graphically representing the color-coded values among three groups. R platform was used to generate heat maps after the transformation of the raw data to z-score.

In total, 18 processed NMR files were subjected to PCA using SimcaP (14.0 version). Pareto scaling was utilized to normalize the data. PCA was performed to analyze intrinsic variation between the samples. In order to analyze the maximum variance between samples supervised approaches such as OPLSD-DA (by using SimcaP) were carried out. The quality of the models was described by R2 and Q2 values. R2 is defined as the proportion of variance in the data explained by the models and indicates the goodness-of-fit, and Q2 is defined as the proportion of variance in the data predictable by the model, and indicates predictability [67]. The model was validated by performing 100 permutations randomly.

Later, metabolic pathway analysis was performed with the help of the Kegg pathway database (https://www.genome.jp/kegg/pathway.html (accessed on 30 November 2021)).

### 4.7. Statistical Analysis

Graphpad prism software (8.44 version), International Business Machine (IBM)—Statistical Package for Social Sciences (SPSS) version 23, SimcaP (14.0 version), and R programming were used for the data analysis. We performed a one-way analysis of variance (ANOVA) and Tukey’s post hoc test of multiple comparisons to check the level of significance of marker metabolites among three varieties. The statistical analysis was performed with a 95% of confidence level and probabilistic value (*p* < 0.05) indicates statistical significance. LDA (Linear Discriminant Analysis) was done to classify the metabolites based on different cotton varieties. Multilayer Perceptron neural network was trained to test for key metabolites for different cotton varieties. Analyses were declared significant for *p*-value < 0.05. 

## 5. Conclusions

^1^H-NMR spectroscopy coupled with multivariate data analysis is a useful method to characterize the metabolic profiles of pest-resistant (FH-142 & MAC-07) and non-pest-resistant (496) cotton varieties. A wide range of bioactive compounds primarily amino acids, carbohydrates, organic acids, terpenoids, fatty acids, and phenylpropanoid derivatives was identified in cotton varieties. We found LDA is a more effective technique than PCA and OPLSDA in classifying cotton varieties on the basis of metabolites. The major discriminating metabolites among cotton varieties are shikimic acid, malic acid, xylulose, maltose, terpinolene, α-ketoglutaric acid, aspartic acid, cinnamic acid, stigmasterol, proline, valine, and fructose. FH-142 (Bt cotton) showed an increase in the concentration of defense-related metabolites, which is probably due to the presence of Cry genes. Furthermore, these metabolites might be responsible for inducing resistance against insect attack in Bt cotton. The bioengineering of defense-related metabolic pathways might be helpful in enhancing the resistance of Bt cotton against insect herbivory.

## Figures and Tables

**Figure 1 molecules-28-01763-f001:**
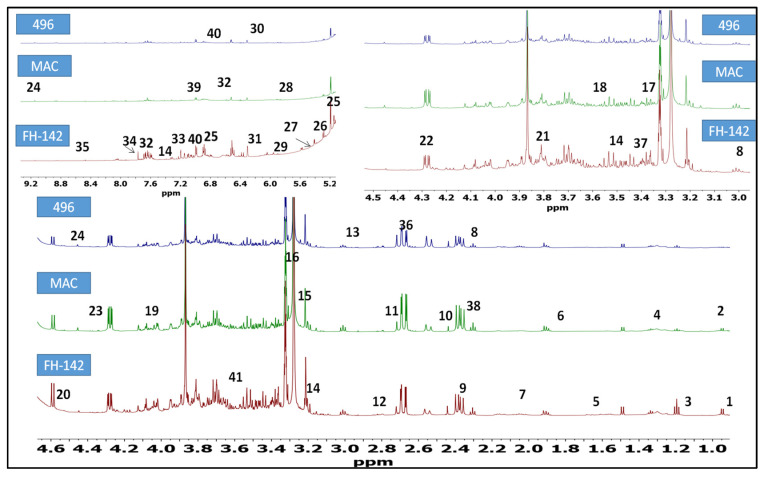
Annotated bioactive constituents in *G. hirsutum* cultivars (NB-01, non-pest-resistant variety); PR-01 (pest-resistant American variety); PR-02 (transgenic pest-resistant variety having Cry proteins, Bt cotton). (1) nonanoic acid, (2) valine, (3) alanine, (4) citrulline, (5) arginine, (6) limonene, (7) linoleic acid, (8) γ-aminobutyric acid; (9) malic acid, (10) succinic acid, (11) terpinolene, (12) di-allylic methylene; (13) asparagine; (14) tryptophan; (15) choline; (16) sycloinositol; (17) proline; (18) glycine; (19) Shikimic acid; (20) arabinose; (21) fructose; (22) xylulose; (23) tartaric acid; (24) trigonelline; (25) e-β-ocimene; (26) stigmasterol; (27) maltose; (28) sucrose; (29) uridine; (30) maleic acid; (31) fumarate; (32) cinnamic acid; (33) tyrosine; (34) dibutyl phthalate; (35) formate; (36) aspartic acid; (37) aconitic acid; (38) α-ketoglutaric acid; (39) chlorogenic acid; (40) ferulic acid; (41) quinic acid.

**Figure 2 molecules-28-01763-f002:**
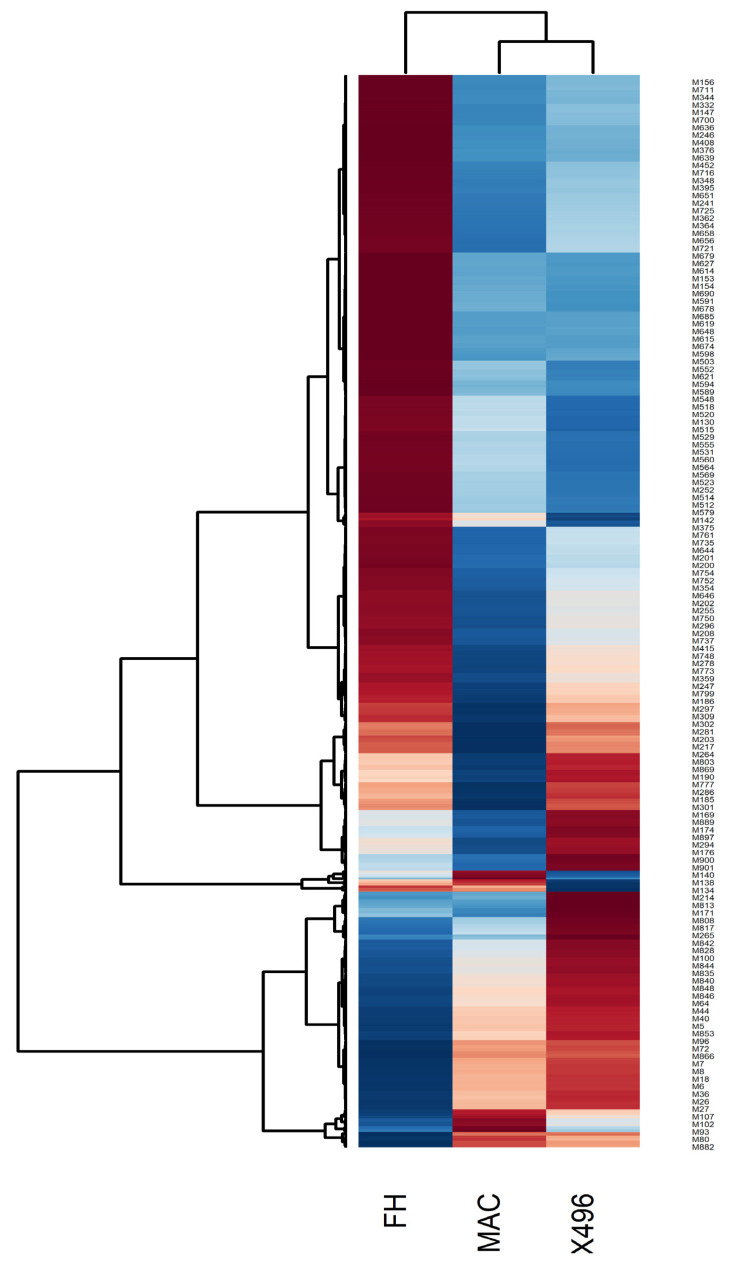
Heat map showing an overall metabolite content in cotton varieties. M376 refers to Arabinose M716 to tyrosine, M348 to tryptophan, M241 to succinic acid, M648 to cinnamic acid, M674 to E-β-ocimene, M515 to maltose, M512 to stigmasterol, M200 to linoleic acid, M646 to fumarate, M296 to GABA, M278 to di-allylic methylene, M186 to limonene respectively. Red color shows an increased expression level while blue color displays decreased expression level. Hierarchical classification was performed to group metabolites from each group. Dendrograms show clustering between metabolites which were closely related based on their estimated Euclidean distance.

**Figure 3 molecules-28-01763-f003:**
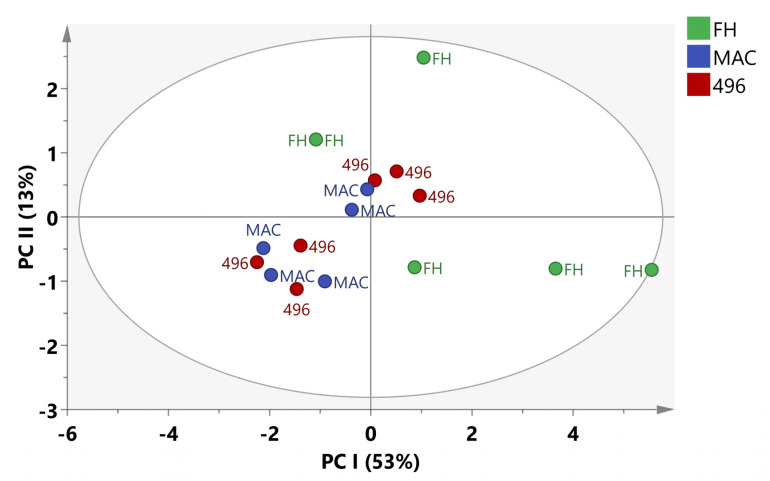
PCA score plot of cotton varieties.

**Figure 4 molecules-28-01763-f004:**
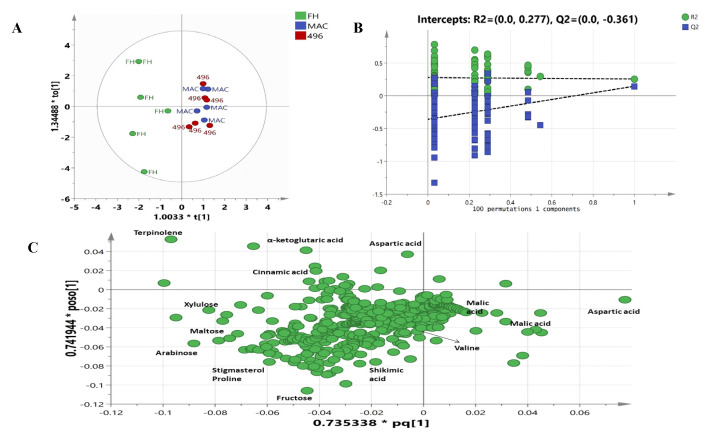
(**A**) OPLSDA score plot of different cotton varieties; (**B**) validation by 100 permutations; (**C**) loading scatter plot of OPLSDA.

**Figure 5 molecules-28-01763-f005:**
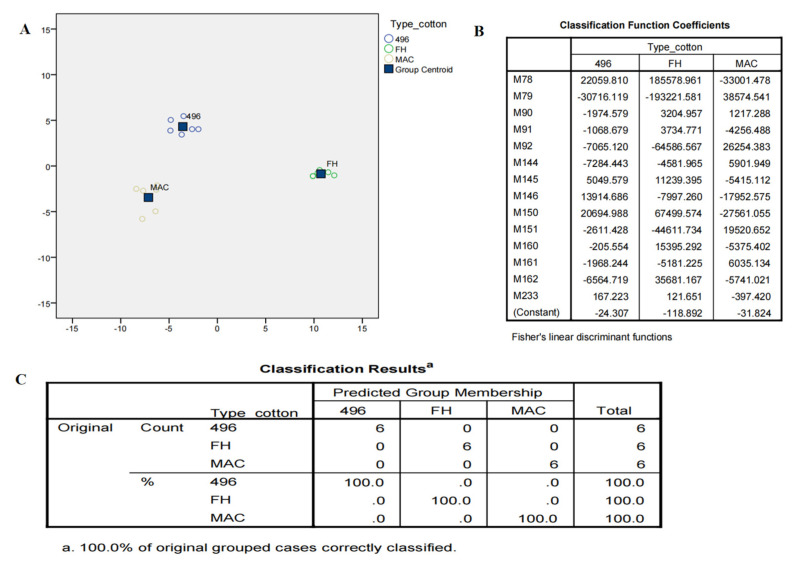
(**A**) Canonical score plot of discriminant analysis showing maximum variance in three varieties of cotton. (**B**) Fisher’s discriminant functions from LDA. M78, M79 are nonanoic acid; M90–M92 are valine; M144–M146 refer to alanine; M150–M151 refer to citrulline; M160–M162 refer to arginine; M233 is malic acid. (**C**) Classification model of the different cotton varieties.

**Figure 6 molecules-28-01763-f006:**
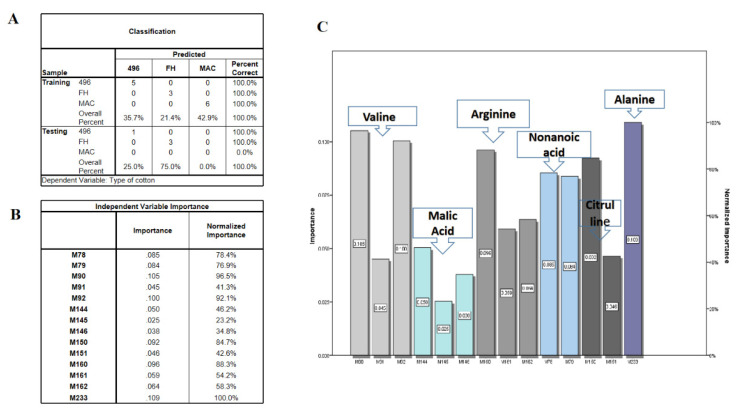
(**A**) Multilayer perceptron neural network results. (**B**) Independent variable importance. M78, M79 are nonanoic acid; M90–M92 are valine; M144–M146 refer to alanine; M150–M151 refer to citrulline; M160–-M162 refer to arginine; M233 is malic acid. (**C**) Fancier versions of importance table.

**Figure 7 molecules-28-01763-f007:**
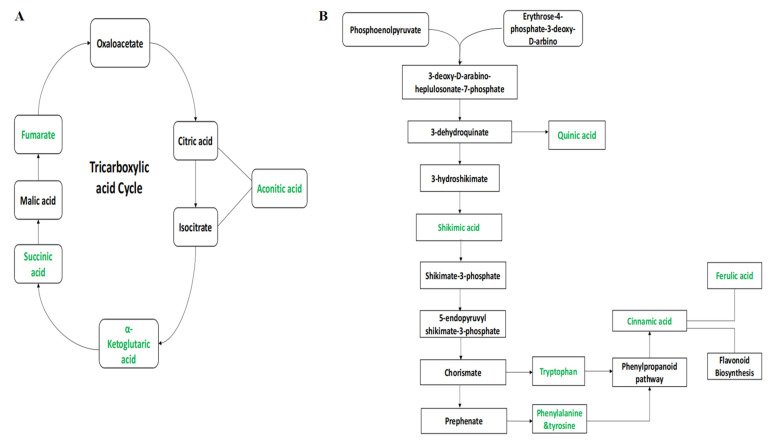
(**A**) Tricarboxylic acid cycle; (**B**) an overview of the secondary metabolites biosynthetic pathways.

**Table 1 molecules-28-01763-t001:** List of identified metabolites from *G. hirsutum*.

Sr. No.	Metabolites	Chemical Shift (ppm)	Multiplicity	Assignment Methods
1	Nonanoic acid	0.82	Doublet	JRES/1D-HNMR
2	Valine	0.95	Doublet	JRES/1D-HNMR
3	Alanine	1.48	Doublet	JRES/1D-HNMR
4	Citrulline	1.56	Multiplet	JRES/1D-HNMR
5	Arginine/myristic acid	1.68	Multiplet	JRES/1D-HNMR
6	Limonene	1.91	Multiplet	JRES/1D-HNMR
7	Linoleic acid	2.06	Multiplet	JRES/1D-HNMR
8	γ-aminobutyric acid	2.30 3.01	triplet triplet	JRES/1D-HNMR
9	Malic acid	2.39	doublet of doublet	JRES/1D-HNMR
10	Succinic acid	2.45	Singlet	JRES/1D-HNMR
11	Terpinolene	2.71	Singlet	JRES/1D-HNMR
12	di-allylic methylene	2.80	Multiplet	JRES/1D-HNMR
13	Asparagine	2.94	Multiplet	JRES/1D-HNMR
14	Tryptophan	3.15	singlet	JRES/1D-HNMR
		3.50	doublet	
		7.53	doublet	
15	Choline	3.20	Singlet	JRES/1D-HNMR
16	Scyloinositol	3.21	Singlet	JRES/1D-HNMR
17	Proline	3.40	triplet of doublet	JRES/1D-HNMR
18	Glycine	3.54	Singlet	JRES/1D-HNMR
19	Shikimic acid	4.01	Multiplet	JRES/1D-HNMR
20	Arabinose	3.82 4.58	doublet of doublet doublet	JRES/1D-HNMR
21	Fructose	3.85	Doublet	JRES/1D-HNMR
22	Xylulose	4.28	doublet of doublet	JRES/1D-HNMR
23	Tartaric acid	4.35	Singlet	JRES/1D-HNMR
24	Trigonelline	4.45	singlet	JRES/1D-HNMR
		9.14	singlet	
25	E-β-ocimene	5.12	singlet	JRES/1D-HNMR
		6.79	doublet of doublet	
26	Stigmasterol	5.15	doublet of doublet	JRES/1D-HNMR
27	Maltose	5.20	Doublet	JRES/1D-HNMR
28	Sucrose	5.45	Doublet	JRES/1D-HNMR
29	Uridine	5.95	Doublet	JRES/1D-HNMR
30	Maleic acid	6.05	Singlet	JRES/1D-HNMR
31	Fumarate	6.50	Singlet	JRES/1D-HNMR
32	Cinnamic acid	6.52	doublet	JRES/1D-HNMR
		7.60	multiplet	
33	Tyrosine	7.20	Doublet	JRES/1D-HNMR
34	Dibutyl phthalate	7.77	Singlet	JRES/1D-HNMR
35	Formate	8.47	Singlet	JRES/1D-HNMR
36	Aspartic acid	2.68	Doublet of doublet	JRES/1D-HNMR
37	Aconitic acid	3.43	Doublet	JRES/1D-HNMR
38	α-Ketoglutaric acid	2.43	triplet	JRES/1D-HNMR
39	Chlorogenic acid	7.12	Doublet of doublet	JRES/1D-HNMR
40	Ferulic acid	7.07 6.91	Doublet of doublet Doublet	JRES/1D-HNMR
41	Quinic acid	3.56	Doublet of doublet	JRES/1D-HNMR

## Data Availability

The data presented in this study are available within the article.

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
