# Peer review of "NMR-Based Metabolomics: A New Paradigm to Unravel Defense-Related Metabolites in Insect-Resistant Cotton Variety through Different Multivariate Data Analysis Approaches"

_molecules, 2023, doi:10.3390/molecules28041763_

Round 1

Reviewer 1 Report

1. the statement on lines 88-89 should explain more. LC-MS should be more powerful than NMR for metabolomics.

2. More wording should be used to introduce the objective of this study.

3. The resolution of Figs 1,2,5,6 should be significantly improved.

4. Table 2 should be regenerated to adjust the size and align with the page, or move to the supporting materials.

5. How authors determined when to harvest the leaf? if they were harvested on the same day and if the metabolites were different under the different growing stages?

6. The authors should explain how to perform the metabolic pathway analysis on lines 454-455. If the author just mapped all the identified metabolites to KEGG database, how to explain if the pathways are significantly changed or not? The number of metabolites each study determined will also impact the results.

Author Response

Dear Editor,

We would like to thank the journal and the reviewers for the valuable and useful comments for our manuscript. 

We have made the following corrections/modifications and additions to our manuscript. Points addressing the specific comments raised by the reviewers, and the detailed responses are listed in the following table:

No.

Comments

Corrections made

Reviewer 1

1

The statement on lines 88-89 should explain more. LC-MS should be more powerful than NMR for metabolomics.

We are thankful to reviewer for the suggestion. Now we have explained in little more detail regarding the use of NMR instead of LCMS in revised version.

2

More wording should be used to introduce the objective of this study.

As per the reviewer’s comment amendments have been made.

3

The resolution of Figs 1,2,5,6 should be significantly improved.

The resolution of all the figures recommended by reviewer to be improved has been enhanced in revised version. The fig. 1 and 2 were merged and fig. 5 and 6 were numbered as fig. 4 and 5 in the revised manuscript.

4

Table 2 should be regenerated to adjust the size and align with the page, or move to the supporting materials.

Table 2 has been moved to the supplementary file

5

How authors determined when to harvest the leaf? if they were harvested on the same day and if the metabolites were different under the different growing stages?

We have harvested the cotton leaves six months after the sowing of seeds. According to the literature, the expression of Bt genes is high during the early stages. Furthermore, it is affected due to its overexpression during developmental stages (Lanquin et al. 2005). Therefore, keeping in view of the above-mentioned factors we decided to harvest it after six months.

6

The authors should explain how to perform the metabolic pathway analysis on lines 454-455. If the author just mapped all the identified metabolites to KEGG database, how to explain if the pathways are significantly changed or not? The number of metabolites each study determined will also impact the results.

We have mapped the metabolic pathways by searching identified metabolites in KEGG database. Then we mapped those pathways in which the identified metabolites were found either up regulated or down regulated.

We have endeavored to the best of our ability to address the points raised by the reviewers. It is hoped that the quality of the manuscript has improved and can now be considered for publication. We would like to thank the editors and reviewers once again for going through our responses.

Sincerely,

Authors

Reviewer 2 Report

In the current manuscript, the authors report the identification of key metabolites responsible for insect resistance in Bt cotton and compare the conventional multivariate analysis methods with deep learning approaches to perform clustering analysis. Metabolite profiling was performed by 1D 1H NMR spectroscopy. In my opinion, the manuscript has some shortcomings addressed below and is not suitable for publication in the present form.

1. There are many errors in grammar and clarity, and a native English speaker should edit the manuscript.

2. In the Introduction section authors discuss on different analytical techniques which are commonly used in plant metabolomics without mentioning hybrid NMR techniques such as LC-NMR and LC-SPE-NMR. In addition, they only mention advantages of NMR spectroscopy, without mentioning the typical disadvantages.

3. Metabolite identification was performed by using 1D 1H NMR spectroscopy. It would be beneficial if the authors would record and present 2D NMR spectra, such as COSY, HSQC and/or HMBC. In addition, they should comment on chemical shift differences between their and literature NMR data. In the Materials and Methods section, they should correct 600 mHz spectrophotometer to 600 MHz spectrometer.

Author Response

Dear Editor,

We would like to thank the journal and the reviewers for the valuable and useful comments for our manuscript. 

We have made the following corrections/modifications and additions to our manuscript. Points addressing the specific comments raised by the reviewers, and the detailed responses are listed in the following table:

No.

Comments

Corrections made

Reviewer 2

1

 There are many errors in grammar and clarity, and a native English speaker should edit the manuscript.

We have amended the grammatical mistakes and improved English language in revised draft of the manuscript.

2

In the Introduction section authors discuss on different analytical techniques which are commonly used in plant metabolomics without mentioning hybrid NMR techniques such as LC-NMR and LC-SPE-NMR. In addition, they only mention advantages of NMR spectroscopy, without mentioning the typical disadvantages.

As per the reviewer’s comment, we have included the use of hybrid NMR techniques like LC-NMR and LC-SPE-NMR in metabolomics, including the disadvantages of NMR compare to other analytical platforms in revised manuscript.

3

Metabolite identification was performed by using 1D 1H NMR spectroscopy. It would be beneficial if the authors would record and present 2D NMR spectra, such as COSY, HSQC and/or HMBC. In addition, they should comment on chemical shift differences between their and literature NMR data. In the Materials and Methods section, they should correct 600 mHz spectrophotometer to 600 MHz spectrometer.

We also recorded the 2D J-resolve of the cotton varieties whose figures have been added as the supplementary file. Additionally the 600 mHz had been corrected in the manuscript to 600 MHz as suggested by the reviewer.

We have endeavored to the best of our ability to address the points raised by the reviewers. It is hoped that the quality of the manuscript has improved and can now be considered for publication. We would like to thank the editors and reviewers once again for going through our responses.

Sincerely,

Authors

Reviewer 3 Report

The authors presented an interesting study in which they were able to identify the metabolites responsible for protecting cotton varieties from insects using modern approaches of multidimensional analysis. The work deserves publication, but some things should be serious revision.

1) "1" in 1H NMR should be written using the superscript. Check the entire text.

2) In tne subsection 2.1.1. the authors mention Table 1 and 2D J-resolved experiment. But I didn't find them in the text. Without this, it is impossible to be sure that the identification of metabolites is carried out correctly.

3) The quality of Figures 1,2, 6 and 8 is low, it needs to be improved.

4) In my opinion, the article is very large. Some points, for example, Table 2 and Figure 9 should be moved to supplementary material. 

5) The reference number 25 was not found in the text.

6) The section "3. Discussion" is overloaded and contains little information concerning the discussion of the data obtained. In this section, the authors pay great attention to information from literary sources, which is usually included in the "Introduction". Thus, section No. 3 should be rewritten.

7) Line 428: "mHz" should be changed on "MHz".

8) Why didn't the authors use an NMR sequence to suppress the water signal? This would greatly facilitate spectrum analysis. In addition, in the section "4.5 Data processing" it is written that "The residual water (δ 4.7-4.8) was excluded from the data", but based on Figure 2 it can be seen that the water signal in the case of FH-142 is much wider (up to 5.5 ppm). Accordingly, this could affect the signals intensities in this area and, accordingly, the results of multidimensional ananlysis.

9) Why did the authors bucketed the spectrum into cells of equal width, and not just upload data in the "ppm-intensity" format?

10) Based on Figure 1, it can be seen that for FH-142, the signals in the range of 2.2-2.8 ppm do not coincide in chemical shift with the signals of other samples. How did the authors solve this problem in the course of multidimensional analysis?

Author Response

Dear Editor,

We would like to thank the journal and the reviewers for the valuable and useful comments for our manuscript. 

We have made the following corrections/modifications and additions to our manuscript. Points addressing the specific comments raised by the reviewers, and the detailed responses are listed in the following table:

No.

Comments

Corrections made

Reviewer 3

1

"1" in 1H NMR should be written using the superscript. Check the entire text.

We are thankful to reviewer for the correction. We have made corrections in the revised manuscript.

2

In the subsection 2.1.1. the authors mention Table 1 and 2D J-resolved experiment. But I didn't find them in the text. Without this, it is impossible to be sure that the identification of metabolites is carried out correctly.

As per the reviewer’s comment, we have added Table 1 in the text of revised draft and 2D J-resolved in supplementary data.

3

The quality of Figures 1,2, 6 and 8 is low, it needs to be improved.

The resolution of all the figures recommended by the reviewer to be improved has been enhanced in revised version. We have merged the figure 1 and 2. Additionally, we have renumbered all the figures in the revised manuscript.

4

In my opinion, the article is very large. Some points, for example, Table 2 and Figure 9 should be moved to supplementary material.

Table 2 and figure 9 have been moved to the supplementary file.

5

The reference number 25 was not found in the text.

We are thankful to the reviewer for pointing out very important mistake. The mentioned reference has been added in the text now.

6

The section "3. Discussion" is overloaded and contains little information concerning the discussion of the data obtained. In this section, the authors pay great attention to information from literary sources, which is usually included in the "Introduction". Thus, section No. 3 should be rewritten.

We are completely agreed with the reviewer that discussion part is overloaded with lot of references from literature. However, in metabolomic study one need to add references from literature in order to justify the metabolomic experimental outcomes. As there is no comparative  study regarding the regulation of metabolites in Bt and non-Bt varieties, therefore, to propose the possible role of metabolites found responsible for separating 3 varieties in MvDA and were might be involved in  inducing resistance in Bt varieties by linking their roles reported in other studies.

7

Line 428: "mHz" should be changed on "MHz".

Correction has been made as per reviewer recommendation.

8

Why didn't the authors use an NMR sequence to suppress the water signal? This would greatly facilitate spectrum analysis. In addition, in the section "4.5 Data processing" it is written that "The residual water (δ 4.7-4.8) was excluded from the data", but based on Figure 2 it can be seen that the water signal in the case of FH-142 is much wider (up to 5.5 ppm). Accordingly, this could affect the signals intensities in this area and, accordingly, the results of multidimensional analysis

In fact in Fig.2, the NMR spectra were zoomed in to show the smaller signals present in the spectra and that could be the reason of water signal seen quite large. But in MvDA, we haven’t changed the signal intensities and later in the software while applying MvDA we excluded the bins having the peak areas of water signal.

9

Why did the authors bucketed the spectrum into cells of equal width, and not just upload data in the "ppm-intensity" format?

Binning is done to digitalize the numeric values for MvDA. This method is common in pant metabolomics to reduce the variation between the samples regarding amount of tissue extracted.

1.     Craig, A. et al. Scaling and normalization effects in NMR spectroscopic metabonomic data sets. Anal. Chem. 78, 2262–2267 (2006). 72.

10

Based on Figure 1, it can be seen that for FH-142, the signals in the range of 2.2-2.8 ppm do not coincide in chemical shift with the signals of other samples. How did the authors solve this problem in the course of multidimensional analysis?

We are thankful to reviewer for highlighting very important point. In fact, the representative spectra of FH-142 shown in Fig. 1 has little shifting in the region pointed out by the reviewer and this very sample was excluded from the MvDA. As it could be seen in PCA and OPLS-DA analysis that FH-142 has one less sample compare to other varieties. By chance, that spectra got included in the Fig. 1 which now has been removed and other representative spectra of FH-142 has been added in revised manuscript.

We have endeavored to the best of our ability to address the points raised by the reviewers. It is hoped that the quality of the manuscript has improved and can now be considered for publication. We would like to thank the editors and reviewers once again for going through our responses.

Sincerely,

Authors

Round 2

Reviewer 2 Report

The authors revised the manuscript according to reviewers comments and the reviewer recommends its acceptance.

Reviewer 3 Report

The authors have made all the relevant edits. I recommend the article for publication.